# Radiation-Induced Hemorrhagic Cystitis in Prostate Cancer Survivors: The Hidden Toll

**DOI:** 10.3390/medicina60111746

**Published:** 2024-10-24

**Authors:** René Gatsinga, Benjamin J. H. Lim, Navin Kumar, Jacinda G. G. Tan, Youquan Li, Michael L. C. Wang, Terence W. K. Tan, Jeffrey K. L. Tuan, Yu Guang Tan, Kenneth Chen, John S. P. Yuen

**Affiliations:** 1Department of Urology, Singapore General Hospital, Singapore 169608, Singapore; gatsinga.rene@sgh.com.sg (R.G.); navin.kumar@mohh.com.sg (N.K.); tan.yu.guang@singhealth.com.sg (Y.G.T.); kenneth.chen@singhealth.com.sg (K.C.); 2Division of Radiation Oncology, National Cancer Center Singapore, Singapore 168583, Singapore; jacinda.tan.g.g@nccs.com.sg (J.G.G.T.); li.youquan@singhealth.com.sg (Y.L.); michael.wang.l.c@singhealth.com.sg (M.L.C.W.); terence.tan.w.k@singhealth.com.sg (T.W.K.T.); jeffrey.tuan.k.l@singhealth.com.sg (J.K.L.T.)

**Keywords:** radiation-induced hemorrhagic cystitis, prostate cancer, radiation therapy, prostate cancer survivorship

## Abstract

*Background and Objectives*: Radiation therapy (RT) plays a crucial role in managing prostate cancer, offering effective disease control and improving survival rates in both localized and recurrent cases. However, RT can lead to hemorrhagic cystitis, a significant late complication resulting in chronic morbidity and other health issues. This study aims to evaluate the real-world incidence of radiation-induced hemorrhagic cystitis requiring surgical intervention. *Materials and Methods*: This retrospective cohort study analyzed data from prostate cancer survivors treated for hematuria at our center between January 2014 and January 2024. Patients were included if cystoscopy identified radiation cystitis as the cause of hematuria. Descriptive statistics were used, and binomial logistic regression analyses with univariate and multivariate analysis were performed to identify risk factors for worse outcomes. *Results*: Fifty-two patients met the inclusion criteria. The estimated cumulative incidence at a median follow-up of 5.3 years was 4.5%. Among the participants, 21.2% required more than two transurethral bladder fulguration (TUBF) procedures, and 38.5% needed more than two hospital admissions for hematuria management. The median time to the first fulguration was 64 months. Blood transfusions were necessary in 53.8% of cases, and 38.5% required hyperbaric oxygen therapy. Ultimately, 5.8% of the patients underwent cystectomy. Univariate analysis identified ischemic heart disease (IHD) and antiplatelet therapy as significant risk factors (OR: 5.17 and 5.18, respectively), along with longer time to first fulguration (OR: 5.02). Multivariate analysis confirmed antiplatelet therapy (OR: 2.8, *p* = 0.05) and time to first TUBF (OR: 1.8, *p* = 0.02) as significant predictors of multiple procedures. *Conclusions*: Radiation cystitis remains a significant burden on prostate cancer survivors. Patients on antithrombotic agents, those with delayed initial presentations, and those who received radiation as salvage therapy are more likely to experience higher morbidity.

## 1. Introduction

Prostate cancer (PC) is the most frequently diagnosed malignancy among men globally, with a cumulative risk of 3.68% by the age of 74 [1]. Along with radical prostatectomy, radiation therapy (RT) of the prostate is a cornerstone in the management of prostate cancer, providing excellent oncological outcomes for both primary treatment of localized disease and salvage therapy for recurrent disease. A recent large prospective study by Hamdy et al. demonstrated that the cancer-specific mortality rate after RT was low and comparable to radical prostatectomy [2]. Hence, when selecting a treatment modality in clinical practice, the side effects profiles are crucial in the decision-making process. Since RT is a nonoperative treatment, it is associated with a significantly lower risk of erectile dysfunction or urinary incontinence in comparison to prostatectomy [3]. However, RT is also associated with various complications due to off-target radiation to tissue in the treated area, even if not cancerous [4]. In the setting of prostate cancer, the scattered radiation most often affects bladder and bowel functions.

One of the more important complications is radiation-induced hemorrhagic cystitis (RHC), which can cause significant long-term morbidity and is, at times, life-threatening [4]. The reported incidence rates in the literature range from 2.8% to 17% [5,6,7,8]. Its pathophysiology is intricate and remains poorly understood. Various mechanisms have been proposed to elucidate its development [9]. Notably, the trauma inflicted on the urothelium, vasculature, and detrusor muscle culminates in the replacement of smooth muscle with fibroblasts, perivascular fibrosis, and vascular ischemia of the bladder wall, leading to critical disruption of the native architecture. These processes begin in the days following RT administration and can continue for many years thereafter [10].

As a consequence of its complex pathophysiology, radiation cystitis can manifest either acutely or as a chronic condition that may emerge many years after treatment. The clinical presentation of radiation cystitis varies widely in severity, ranging from brief, mild lower urinary tract symptoms to severe, debilitating conditions. In more serious cases, radiation cystitis can lead to significant morbidity, necessitating repeated procedures, hospital admissions, and, at times, major surgery [7,8]. In fact, Ma et Al reported that 7.2% of all emergency urological admissions were related to complications of radiation therapy [11]. This heterogeneity complicates the accurate estimation of its true incidence at a population level, as very few prospective studies have a follow-up duration long enough to capture the chronic presentations. 

In this study, we aim to evaluate the real-world incidence rate and burden of radiation-induced hemorrhagic cystitis requiring surgical intervention, classified as Common Terminology Criteria for Adverse Events (CTCAE) grade 3 or higher, following radical radiotherapy for prostate cancer treatment at a high-volume tertiary care center in Southeast Asia [12].

## 2. Materials and Methods

### 2.1. Data Source

This retrospective observational cohort study uses real-world data from a prospectively maintained database of patients undergoing treatment for prostate cancer in a large tertiary academic medical institution in Southeast Asia. We identified men who underwent endoscopic intervention for hematuria at our center between January 2014 and July 2024. We reviewed their health records to assess personal and clinical characteristics. Patients were included in the study if their cystoscopy intervention report identified radiation cystitis as the cause of hematuria and if they had previously undergone radiotherapy for prostate cancer. The diagnosis of RHC was defined cystoscopically based on the classical findings of mucosal pallor, diffuse telangiectasia, and petechiae, with or without ulcerations, fitting the pathological description by Fajardo et al. [13]. We excluded patients whose hematuria was attributed to other causes, those who had received radiotherapy to the pelvis for other pathologies, or those with a recent diagnosis of urothelial malignancy. The data collected for each participant included age, Charlson Comorbidity Index (CCI), body mass index (BMI), chronic medication, date of PC diagnosis and treatment, total dose of RT prescribed, date of first endoscopic intervention, and details of treatment received for radiation cystitis. Upon establishing the range of RT treatment dates for all included participants, we subsequently retrieved the total number of prostate cancer patients who received RT at our center during the same period from the department archive. These data were then utilized as denominators for the incidence rate estimate calculations. The study outcomes were to assess the incidence of CTCAE grade 3 or higher RHC needing surgical intervention, the number of interventions required, and the overall morbidity and mortality arising from RHC. Institutional Review Board ethics approval was obtained (CRIB Ref 2009/1053/D), and the study was conducted according to the principles of the Helsinki Declaration and Good Clinical Practice guidelines. 

### 2.2. Statistical Analyses

We used standard descriptive statistics to summarize the data. Categorical variables were summarized as frequencies (*n*), and proportions (%), and continuous variables were summarized as the median and interquartile range (IQR). Listwise deletion was used to handle missing data. Binomial logistic regression was employed to conduct univariate and multivariate analyses to identify risk factors associated with worse outcomes. The annual total number of patients who received radiotherapy for prostate cancer during the study period was obtained from department records and used as the denominator to estimate incidence rates. R software version 4.2.1 was used to compute statistical analyses and generate graphic illustrations [9].

## 3. Results

### 3.1. Cohort Characteristics

Between January 2014 and July 2024, 258 patients underwent cystoscopy and transurethral bladder fulguration (TUBF) at our center. Fifty-four of them fulfilled the study criteria, but two were excluded due to missing data. Fifty-two participants were eventually included in the study. Table 1 summarizes the cohort characteristics. The median age of the patients was 70 years, with an IQR of 65 to 73 years. The median PSA level was 14.4, ranging from 8.3 to 24.1. Regarding Gleason scores, 7.7% of the patients had a score of 6, 53.8% had a score of 7, 13.5% had a score of 8, another 13.5% had a score of 9, 5.8% had a score of 10, and 3.8% had tumors of neuroendocrine histology with no Gleason grading applicable. Metastatic disease was present in 19.2% of the patients. The median BMI was 23.1, with an IQR of 20.4 to 25.5. Ischemic heart disease (IHD) was present in 55.8% of the patients, while 75% had hypertension, and 42.3% had diabetes mellitus. An age-adjusted CCI greater than 2 was observed in 50% of the patients. Regarding ASA scores, 67.3% of the patients had a score of 2, and 32.7% had a score of 3. Over half of the patients (53.7%) were on antithrombotic therapy. The dates of PC diagnosis ranged from November 2000 to October 2022, and the dates of RT ranged from November 2000 to December 2022. For the majority of patients, 78.8%, the indication for radiotherapy was primary treatment, while 21.2% had it as salvage. The doses of RT prescribed ranged from 36.25 Gy to 78 Gy. The primary radiation oncologist determined the RT doses for each patient individually. However, the most commonly prescribed dose was 74 Gy, which was administered to 46% of the participants.

Among the 52 patients, 21.2% needed more than two transurethral bladder fulguration procedures, while 38.5% required more than two hospital admissions to manage hematuria. The median time to the first fulguration was 64 months (IQR: 33–97 months), with the dates of initial interventions ranging from July 2009 to January 2024. Blood transfusions were necessary for 53.8% of the patients, and 38.5% required hyperbaric oxygen therapy. Ultimately, 5.8% of the patients underwent definitive surgical treatment with cystectomy.

### 3.2. Estimated Incidence Rate

During the period from November 2000 to October 2022, when the participants of this cohort were first diagnosed and treated for PC, our center administered curative radiotherapy to 2533 prostate cancer patients, averaging 115 patients per year. Figure 1 illustrates that a small proportion of these patients later developed RHC.

As shown in Figure 2, the annual incidence of CTCAE grade ≥ 3 RHC observed at our center between 2014 and 2024 varied from 2 to 8 cases per year, with an average of 5.2 cases annually.

Although true incidence rates are challenging to estimate without a large prospective cohort, these data suggests that a center treating an annual average of 115 prostate cancer patients with curative RT, after a median follow-up duration of 64 months, might expect to observe approximately 5.2 new cases of grade 3 or higher RHC each year. This would amount to an extrapolated 4.5% cumulative incidence after a median follow-up of 5.3 years.

### 3.3. Outcome Predictors

#### 3.3.1. Factors Associated with the Need for Multiple TUBF Procedures

The characteristics associated with the risk of multiple procedures are presented in Table 2. In univariate analysis, ischemic heart disease (IHD) and antiplatelet therapy were significantly associated with an increased risk, with odds ratios (OR) of 5.17 (95% CI: 1.04–26.8) and 5.18 (95% CI: 1.08–26.8), respectively. A longer time to the first diathermy was also significant, with an OR of 5.02 (95% CI: 1.12–22.6).

In multivariate analysis, antiplatelet therapy remained a significant predictor of the need for multiple procedures, with an adjusted OR of 2.8 (95% CI: 1.03–18.1) and a *p*-value of 0.05. Time to the first TUBF also remained significant, with an adjusted OR of 1.8 (95% CI: 1.4–24.6) and a *p*-value of 0.02. IHD was considered a confounder in multivariate analysis. Other factors, notably age, age-adjusted CCI, BMI, hypertension, diabetes, salvage radiotherapy, and RT dose, did not show significant associations with the need for multiple procedures in either univariate or multivariate analysis.

#### 3.3.2. Factors Associated with the Need for Multiple Admissions

We explored factors associated with the need for multiple (more than two) hospital admissions, as presented in Table 3. In univariate analysis, younger age was significantly associated with a reduced likelihood of multiple admissions, with an OR of 0.46 (95% CI: 0.23–0.91) and a *p*-value of 0.02. Additionally, a higher age-adjusted Charlson Comorbidity Index (CCI) was associated with an increased risk of recurrent admissions, with an OR of 1.94 (95% CI: 1.03–3.68) and a *p*-value of 0.04. Salvage radiotherapy (RT) also showed a significant association with an increased need for multiple admissions, with an OR of 4.4 (95% CI: 1.00–20.48) and a *p*-value of 0.05.

In multivariate analysis, salvage RT remained a significant predictor of multiple admissions, with an adjusted OR of 8.3 (95% CI: 1.03–47.4) and a *p*-value of 0.04. However, the significance of age diminished, with a *p*-value of 0.06, and age-adjusted CCI lost its significance in the multivariate model, with a *p*-value of 0.30. Other factors, including BMI, hypertension, diabetes, ischemic heart disease, antithrombotic therapy, radiotherapy dose, and time to first TUBF, did not show significant associations with the need for multiple admissions in either univariate or multivariate analysis.

### 3.4. Outcomes After Definitive Surgery

Three patients in our cohort required a cystectomy to control the refractory recurrent hematuria. Their detailed characteristics are summarized in Table 4. 

The first patient, aged 66, had a history of ischemic heart disease on antithrombotic therapy, type 2 diabetes mellitus, and hypertension. Initially, he underwent a robot-assisted radical prostatectomy but experienced a recurrence 1 year later, necessitating salvage RT. He developed RHC symptoms and had a first TUBF 7 years and 10 months from the RT treatment date. Over the next 2 years and 2 months, he underwent six TUBF procedures prior to proceeding with definitive cystectomy. His recovery was complicated by a pelvic abscess and a colocutaneous fistula, which required a diverting colostomy. He remains free of prostate cancer at the time of reporting, with an undetectable serum PSA. 

The second, aged 72, had hypertension, hyperlipidemia, and atrial fibrillation on antithrombotic therapy. He received RT as primary treatment and developed RHC symptoms, necessitating a first TUBF 3 years and 9 months later. Over the following 3 years and 5 months, he had 11 hospital admissions for hematuria treatment. He eventually underwent a cystectomy, which was complicated by a small pelvic abscess treated with a long course of antibiotics. At the conclusion of this study, he has no PC recurrence and a stable serum PSA value of 0.57 ng/dL.

The third patient, aged 82, had interstitial lung disease, hypertension, and chronic kidney disease. He received RT for the primary treatment of prostate cancer. He subsequently developed RHC, requiring a first TUBF 4 years and 6 months after the initial RT treatment. Over 1 year and 5 months, he required eight hospital admissions and three TUBF procedures before undergoing cystectomy. Unfortunately, he experienced a flare of interstitial lung disease postoperatively, leading to respiratory failure and demise on postoperative day 29.

## 4. Discussion

This study highlights the significant burden of chronic radiation therapy complications among prostate cancer survivors. Approximately 4.5% of patients in our series developed radiation cystitis requiring TUBF, with over half needing blood transfusions, 37% requiring two or more hospital admissions to manage their hematuria, and 5% eventually needing a cystectomy. These figures likely underestimate the true incidence, as they do not account for patients who may have sought care at other centers. Since this study only captured cases requiring surgical intervention, the actual burden is probably much higher and more challenging to quantify. Traditionally, RHC was thought to occur in less than 5% of patients who received pelvic RT [9,14,15]. However, some recent reports suggest that, with longer follow-up, the burden is much more significant. For instance, David et al. in Australia estimate that the RHC burden at the 60 months follow-up reaches as high as 33% [7]. With a follow-up duration reaching 20 years, our findings corroborate these higher estimates. 

We were able to identify statistically significant risk factors for more severe forms of RHC. Among the patients requiring surgical intervention, those on antithrombotic agents were more likely to necessitate repeated transurethral fulguration procedures, a predictable outcome given the nature of such therapy. Sanguedolce et al., in Spain, reported similar findings. In their recent study, they identified antithrombotic therapy as an independent risk factor for developing RHC in their cohort [6]. We also showed that patients who required intervention at a time that was delayed from the initial RT date were significantly more likely to need multiple repeated procedures. This observation has not been reported in studies elsewhere and may indicate that patients with a delayed presentation exhibit a more severe and refractory form of the disease. 

Patients who received RT in a salvage setting were also more likely to require multiple hospital admissions to manage their hematuria. We hypothesize that this may be attributable to the higher cumulative dose of radiation in the salvage setting, particularly if RT was also administered as the primary treatment. On the other hand, for patients who have undergone prostatectomy as the primary treatment, the resultant altered bladder anatomy might predispose them to more severe cystitis. Supporting this, Makino et al. reported increased rates of cystitis in the salvage RT setting [16]. They posit that bladder dysfunction and higher incontinence rates following prostate cancer treatment impede the bladder’s ability to distend fully, thereby exposing more bladder tissue to radiation. This complex interplay warrants further dedicated research to elucidate the underlying mechanisms.

Interestingly, previously established risk factors such as higher BMI and older age [5,17] did not reach statistical significance in the multivariate analysis of our cohort. This may be attributed to the skewed nature of our cohort, which consists of CTCAE grade 3 or higher RHC at baseline. Consequently, detecting significant differences in such a cohort requires an appropriately powered sample size. 

Currently, there are no established guidelines on the management options for RHC. Although numerous therapies have been described in the literature as effective in controlling RHC symptoms in both acute and chronic settings, many of these reports are single-center series, retrospective reviews, or expert opinions. Consequently, high-quality data to guide treatment are still lacking. In our cohort, the treatment patterns exhibit variability but are comparable to those documented in studies from other geographical regions. For instance, in a large cohort from the United States, Bologna et al. report that HBOT was the most commonly used treatment following TUBF, with 16% of their patients receiving this therapy [5]. Similarly, after managing acute hematuria episodes, HBOT was the most frequently prescribed treatment to our patients, with 38.5% receiving it to reduce the risk of recurrence. We attribute the relatively higher proportion of patients requiring treatment in our cohort to the fact that we only included patients with RHC of grade CTCAE grade 3 or higher. Multiple studies have reported significant improvement in RHC symptoms with HBOT, establishing this treatment as a mainstay modality for managing chronic RHC [18,19]. However, in our cohort of patients with moderate-to-severe cystitis, no significant reduction in the risk of requiring multiple surgical procedures or hospital admissions was detected among patients who received HBOT. 

Ultimately, 5.8% (3 out of 52) of the patients required complete urinary diversion through cystectomy and ileal conduit. Although the number of patients needing cystectomy due to intractable RHC bleeding is small, this surgery is associated with considerable mortality and morbidity rates, as reported by Linder et al. [20]. In our series, the morbidity rate following cystectomy was markedly high, with one patient succumbing within 30 days of surgery and the other two experiencing significant complications of Clavien–Dindo grades II and IIIb. Due to the extremely small size of our cohort, these figures are challenging to compare directly with those reported in the literature. In a considerably larger series, Linder et al. reported severe complications (Clavien–Dindo grades III to V) in 42% of their patients, with a 90-day mortality rate of 16% [20]. More recently, a series from another high-volume center by Tachibana et al. reported notably lower, yet significant, severe complication rates of 13% following cystectomy [21]. The elevated morbidity associated with definitive cystectomy obviates its role as a last-resort intervention.

Recent advances in radiation therapy are allowing for more controlled delivery and reduced adverse effects. For example, intensity-modulated radiation therapy employs a computer program to vary the radiation intensity delivered to specific areas of the prostate through multiple fixed-angle beams, significantly reducing off-target radiation and associated adverse effects [22]. Image-guided radiation therapy enhances accuracy by pairing irradiation with real-time or recently obtained images of the target organ, often using computed tomography (CT) or magnetic resonance imaging (MRI) [23,24]. Most recently, proton-beam therapy, which uses protons instead of conventional photons to irradiate tissues, has shown promising results. The physical properties of charged particles inherently reduce the amount of excess radiation delivered to patients compared to conventional photon radiation therapy. Many studies have reported the promising short-term safety and efficacy of proton-beam therapy [25,26]. Some groups are starting to report favorable long-term outcomes [27] and efforts are ongoing to make this technology more accessible [25].

Our study is not without limitations. First, its retrospective design is prone to various biases. However, given that RHC can manifest more than a decade after treatment, conducting prospective trials with adequate follow-up durations would require prohibitive resources. Second, we could only include patients who returned to our center for RHC treatment. This limitation likely led to an underestimation of the true incidence due to the inevitable loss of follow-up commonly observed in real-world settings, especially for such a chronic condition. Third, we could not obtain data pertaining to the quality of life of included participants despite this being of primordial importance among cancer survivors. Last, the data available to us was limited to patients who developed cystitis necessitating surgical intervention, thereby restricting our capacity to identify risk factors for the onset of the disease. Nonetheless, within the subgroup of patients presenting with CTCAE grade 3 or higher cystitis, we were able to ascertain predictors of more severe illness.

## 5. Conclusions

Radiation cystitis constitutes a significant burden among prostate cancer survivors and can result in chronic morbidity. Among patients who require surgical intervention, our findings suggest that those who take antithrombotic agents and have a delayed first presentation are likely to require multiple repeated procedures. Patients who received radiation as salvage therapy were more likely to require multiple repeated hospital admissions. The true incidence of this condition is hard to quantify, but as men continue to live longer, radiation-induced hemorrhagic cystitis is poised to be a weighty threat among prostate cancer survivors.

## Figures and Tables

**Figure 1 medicina-60-01746-f001:**
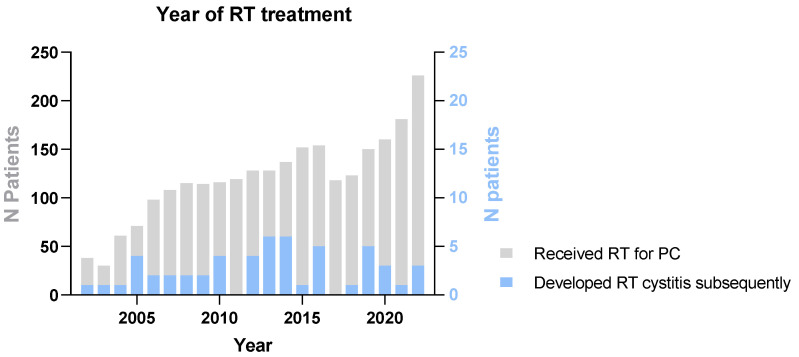
Number of patients treated for prostate cancer per year.

**Figure 2 medicina-60-01746-f002:**
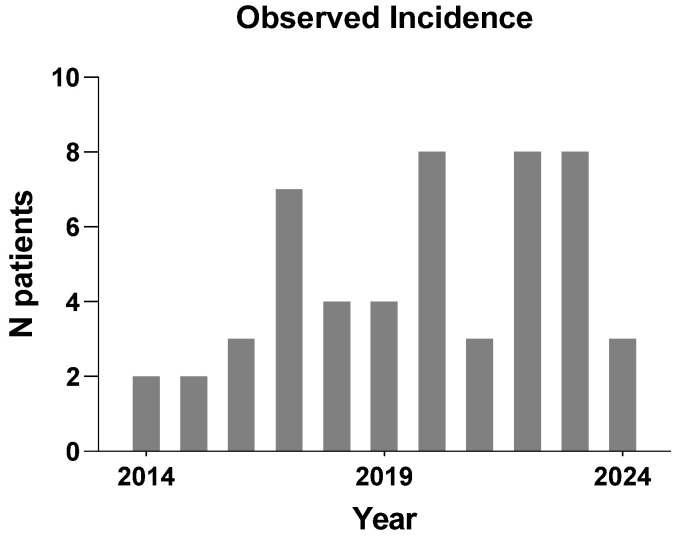
New cases of RHC requiring surgical intervention annually.

**Table 1 medicina-60-01746-t001:** Patient demographics and characteristics.

Characteristics	Median/N (IQR/%)
Age (years)	70 (65–73)
PSA (ng/mL)	14.4 (8.3–24.1)
Gleason sum	
6	4 (7.7%)
7	28 (53.8%)
8	8 (15.4%)
9	7 (13.5%)
10	3 (5.8%)
Neuroendocrine	2 (2.8%)
Metastatic disease	10 (19.2%)
BMI	23.1 (20.4–25.5)
IHD	29 (55.8%)
Hypertension	39 (75%)
Diabetes mellitus	22 (42.3%)
Age-adjusted CCI > 2	26 (50%)
ASA score	
2	35 (67.3%)
3	17 (32.7%)
On anticoagulation or antiplatelets	27 (51.9%)
Primary RT	41 (78.8%)
Salvage RT	11 (21.2%)
Total dose received (Gy)	74 (66–74)
Time to first TUBF (months)	64 (33–97)
Hyperbaric Oxygen Therapy	20 (38.5%)
Required blood transfusion	28 (53.8%)
Required repeated (>2) transurethral fulguration	11 (21.2%)
Required repeated (>2) hospitalization	20 (38.5%)
Radical cystectomy	3 (5.8%)

ASA: American Society of Anesthesiologists; BMI: body mass index; CCI: Charlson Comorbidity Indexl; Gy: Grey unit; IHD: ischemic heart disease; PSA: prostate-specific antigen; RT: radiation therapy; and TUBF: transurethral bladder fulguration.

**Table 2 medicina-60-01746-t002:** Factors associated with the need for more than two TUBF procedures.

Patient Characteristics	Univariate *p*-Value	OR and 95% CI	Multivariate *p*-Value	OR and 95% CI
Age	0.98		0.99	
Age-adjusted CCI	0.69		0.56	
BMI	0.67		0.51	
Hypertension	0.15		0.29	
Diabetes	0.48		0.81	
IHD	**0.05**	**5.17 (1.04–26.8)**	-	confounder
Antiplatelets	**0.05**	**5.18 (1.08–26.8)**	**0.05**	**2.8 (1.03–18.1)**
Salvage RT	0.73		0.49	
RT dose > 70 Gy	0.8		0.99	
Time to first TUBF	**0.03**	**5.02 (1.12–22.6)**	**0.02**	**1.8 (1.4–24.6)**

BMI: body mass index; CCI: Charlson Comorbidity Index; CI: confidence interval; Gy: Grey unit; IHD: ischemic heart disease; OR: odds ratio; RT: radiation therapy; and TUBF: transurethral bladder fulguration; bold values indicate statistical significance.

**Table 3 medicina-60-01746-t003:** Factors associated with more than two hospital admissions.

Patient Characteristics	Univariate *p*-Value	OR and 95% CI	Multivariate *p*-Value	OR and 95% CI
Age	**0.02**	**0.46 (0.23–0.91)**	0.06	
Age-adjusted CCI	**0.04**	**1.94 (1.03–3.68)**	0.30	
BMI	0.64		0.58	
Hypertension	0.68		0.62	
Diabetes	0.82		0.45	
IHD	0.35		0.70	
Antithrombotics	0.13		0.22	
Salvage RT	**0.05**	**4.4 (1.00–20.48)**	**0.04**	**8.3 (1.03–47.4)**
RT dose	0.8		0.99	
Time to first TUBF	0.76		0.77	

BMI: body mass index; CCI: Charlson Comorbidity Index; CI: confidence interval; IHD: ischemic heart disease; OR: odds ratio; RT: radiation therapy; and TUBF: transurethral bladder fulguration; bold values indicate statistical significance.

**Table 4 medicina-60-01746-t004:** Characteristics of patients who underwent cystectomy.

Characteristics	Patient 1	Patient 2	Patient 3
Age	66	72	82
Comorbid conditions	Ischemic heart disease Type 2 diabetes mellitus Hypertension	Hypertension Hyperlipidemia	Atrial fibrillation Interstitial lung disease Hypertension Chronic kidney disease
Initial diagnosis	Gleason 3 + 4, T3aN0M0	Gleason 4 + 3, T2bN0M0	Gleason 3 + 4 T3aN0M0
Treatment	Robot-assisted radical prostatectomy Salvage RT (1 year post prostatectomy) IMRT 66 grey	Primary RT IGRT 74 grey	Primary RT IGRT 66 grey
Time to first TUBF	7 years and 10 months	3 years and 9 months	4 years 6 months
Time from first TUBF to Cystectomy	2 years and 2 months	3 years and 5 months	1 year 5 months
Number of TUBF prior to cystectomy	6	4	3
Number of hospital admissions prior to cystectomy	4	11	8
Mode of urinary diversion	Ileal Conduit	Ileal Conduit	Ileal conduit
Complications	Clavien–Dindo IIIB - Pelvic abscess requiring transurethral drainage - Colocutaneous fistula requiring diverting colostomy	Clavien–Dindo I - Pelvic abscess requiring prolonged antibiotics course	Clavien–Dindo V - Respiratory failure requiring intubation - Demise on post-op day 29
Mortality	No	No	Yes
Cause of death	NA	NA	Interstitial lung disease flare

IGRT: image-guided radiation therapy; IMRT: intensity-modulated radiation therapy; RT: radiation therapy; and TUBF: transurethral bladder fulguration; NA: not applicable.

## Data Availability

The data presented in this study are available upon request from the corresponding author to protect patients’ confidentiality.

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
