# Peer review of "Radiation-Induced Hemorrhagic Cystitis in Prostate Cancer Survivors: The Hidden Toll"

_medicina, 2024, doi:10.3390/medicina60111746_

Round 1
Reviewer 1 Report
Comments and Suggestions for Authors
The authors need to complimented for a very large retrospective review of the radiation-induced hemorrhagic cystitis in prostate cancer survivors.
This is a good compilation of data.
Few comments:
1. Table 1: How many patients formed a part of this study. In abstract you have mentioned 52 patients, but Gleason’s grade is available only for 50. The total of ASA grade is for 54 while the total of primary and salvage RT is 53.
2. Even the percentage calculation does not seem correct; kindly recheck.
3. Line 211: 3 out of 52 is 5.77%
4. Is there an error; the dates of radiation seems to have pre-dated the time of diagnosis (line 99-101)
5. Sentence on line 89 is incomplete.
6. Line 138: spelling of notably
7. Line 212: Instead of ileal conduit ureterostomy, just ileal conduit is good.
Comments on the Quality of English LanguageFew spelling and typo errors
Author Response
Dear Reviewer,
Thank you for your thorough review of our manuscript. We appreciate the valuable feedback and have addressed each of your comments as detailed below
We sincerely appreciate your careful consideration and constructive feedback, which have significantly improved the quality of our manuscript.
We hope that the revised manuscript meets your expectations.
Thank you for your time and effort.
Comment 1. Table 1: How many patients formed a part of this study. In abstract you have mentioned 52 patients, but Gleason’s grade is available only for 50. The total of ASA grade is for 54 while the total of primary and salvage RT is 53.
Thank you for pointing out this mistake. After listwise deletion of 2 participants due to missing data, 52 patients were included in the study. We have corrected the manuscript (Table 1, Lines 111-126) to ensure that the reflected numbers are coherent.
Comment 2. Even the percentage calculation does not seem correct; kindly recheck.
Thank you for pointing this out. We have corrected the values in Table 1 to ensure they tally up accurately. The percentages have been recalculated, and the data now accurately reflects the number of patients included in the study.
Comment 3. Line 211: 3 out of 52 is 5.77%
Thank you for your meticulous review. We have made the necessary corrections as recommended (line 271).
Comment 4: Is there an error; the dates of radiation seem to have pre-dated the time of diagnosis (lines 99-101)
Thank you for your observation. We have reviewed the dates and corrected them accordingly. The first date of diagnosis should indeed be November 2000. The manuscript has been updated to reflect this accurate timeline (Lines124-125).
Comment 5: Sentence on line 89 is incomplete.
Thank you. We have addressed the incomplete sentence on line 89 (now line 111) to ensure it is now complete and coherent.
Comment 6: Line 138: spelling of "notably".
We have corrected the typographical error.
Comment 7. Line 212: Instead of ileal conduit ureterostomy, just ileal conduit is good.
Thank you for the suggestion. We have corrected the sentence accordingly.
Reviewer 2 Report
Comments and Suggestions for Authors
General comment
The manuscript entitled “Radiation-Induced Hemorrhagic Cystitis in Prostate Cancer Survivors: The Hidden Toll” provides valuable insights into the incidence and management of radiation-induced hemorrhagic cystitis in prostate cancer survivors, particularly in a Southeast Asian setting. The study’s strength lies in its large sample size and robust statistical analysis. However, there are several areas that could be improved, including the clarity and fluency of the manuscript, the discussion of limitations and the inclusion of a more comprehensive review of the literature. In detail:
INTRODUCTION
The introduction is somewhat brief and could be expanded providing more epidemiological data regarding PCa and the use of radiotherapy in this setting.
Additionally, albeit the citation of Ma et al.'s study on emergency urological admissions is a valuable addition, a more detailed discussion of the underlying pathophysiology of RHC and potential preventive measures could be a nice addition.
About RHC treatment also cite less invasive alternatives such as: 10.3390/jcm13164724 and 10.3390/nu15163573
MATERIAL AND METHODS
The study could benefit from a more detailed description of the institutional practices for radiation therapy and post-treatment follow-up protocols, as these may influence the incidence and severity of RHC.
The use of a retrospective cohort design is a limitation inherent to the study, and this should be clearly stated. Additionally, while the study mentions that missing data were handled via listwise deletion, the potential impact of this method on the results should be further discussed.
RESULTS
Some tables and figures could be more clearly labeled.
A more detailed discussion on the range of RT doses and their correlation with the incidence of RHC would be beneficial.
DISCUSSION
The discussion would benefit from more detailed comparisons with other similar studies, particularly those conducted in different geographic or institutional settings.
As previously mentioned, a more thorough exploration of the retrospective design and its impact on data completeness and potential bias is warranted.
Comments on the Quality of English LanguageCheck grammar and ensure the fluency and the clarity of the text along the manuscript.
Author Response
Dear Reviewer,
Thank you for your thorough review of our manuscript. We appreciate the valuable feedback and have addressed each of your comments as detailed below
We sincerely appreciate your careful consideration and constructive feedback, which have significantly improved the quality of our manuscript.
We hope that the revised manuscript meets your expectations.
Thank you for your time and effort.
INTRODUCTION
Comment 1 : The introduction is somewhat brief and could be expanded providing more epidemiological data regarding PCa and the use of radiotherapy in this setting.
Thank you for your advice, we have expanded the introduction to elaborate on the epidemiology of prostate cancer and the use of radiotherapy (lines 35-47).
Comment 2: Additionally, albeit the citation of Ma et al.'s study on emergency urological admissions is a valuable addition, a more detailed discussion of the underlying pathophysiology of RHC and potential preventive measures could be a nice addition. About RHC treatment also cite less invasive alternatives such as: 10.3390/jcm13164724 and 10.3390/nu15163573.
We are really grateful for your insight. We have amended the manuscript to add a description of the underlying pathophysiology of RHC in the introduction at lines 48-62, and also discussed the preventive and curative measures used in lines 271-283. We have included the suggested citation in line 268.
MATERIAL AND METHODS
Comment 3: The study could benefit from a more detailed description of the institutional practices for radiation therapy and post-treatment follow-up protocols, as these may influence the incidence and severity of RHC.
Unfortunately, as RHC is still a relatively rare condition, we do not have a standardized protocol to screen for or manage RHC after radiation therapy. After prostate cancer treatment, patients remain on serum PSA surveillance for a minimum of 10 years. Due to our geographical condition, being located in a high-density city, our patients are usually able to return to us for care at any point even after the surveillance period has elapsed.
Comment 4: The use of a retrospective cohort design is a limitation inherent to the study, and this should be clearly stated. Additionally, while the study mentions that missing data were handled via listwise deletion, the potential impact of this method on the results should be further discussed.
Thank you for pointing this out. We have amended the manuscript to clarify that the design of the study was restrospective, although the RHC diagnoses were made by clinicians and documented prospectively (lines 78-83). We also clarified that the data collected was retrospectively matched against the total number from the department statistics (lines 89-92)
RESULTS
Comment 5: Some tables and figures could be more clearly labeled.
Thank you very much. We have amended the tables and figure for clarity and comprehensibility
Comment 6: A more detailed discussion on the range of RT doses and their correlation with the incidence of RHC would be beneficial.
We agree that the complications of RT and RHC specifically are known to be dose-dependent. However, as this study only looks at patients treated for prostate cancer, they all received relatively similar doses of RT, and the range was too narrow for us to detect a potential correlation with the incidence of RHC. We have added more details about the doses of RT prescribed to our cohort in lines 126-129.
DISCUSSION
Comment 7: The discussion would benefit from more detailed comparisons with other similar studies, particularly those conducted in different geographic or institutional settings.
Thank you very much for this input; we have amended the discussion to compare our findings to those of multiple studies for different geographical areas in lines 222-283.
Comment 8: As previously mentioned, a more thorough exploration of the retrospective design and its impact on data completeness and potential bias is warranted.
We have clarified in lines 111-112 how the handling of missing data by listwise deletion affected the results of this retrospective cohort study. Only 2 participants were excluded. Given the nature of RHC and it’s very long onset time, prospective data is very difficult to obtain. Consequently, the vast majority of reported studies on RHC, which are quoted in this manuscript, are of retrospective design. We believe this study has a risk of bias that is similar to the standard for real-world retrospective study.
Comments on the Quality of English Language
Comment 9: Check grammar and ensure the fluency and the clarity of the text along the manuscript
Thank you very much for your review. We have edited the manuscript for grammar and readability
Round 2
Reviewer 2 Report
Comments and Suggestions for Authors
The authors improved the manuscript according to previous suggestions. However, check this point again:
Additionally, albeit the citation of Ma et al.'s study on emergency urological admissions is a valuable addition, a more detailed discussion of the underlying pathophysiology of RHC and potential preventive measures could be a nice addition. About RHC treatment also cite less invasive alternatives such as: 10.3390/jcm13164724 and https://doi.org/10.3390/nu15163573
Comments on the Quality of English LanguageNone
Author Response
Dear Reviewer,
Thank you for your valuable feedback on our manuscript. We have carefully considered your suggestion regarding the inclusion of a more detailed discussion of the underlying pathophysiology of RHC. Accordingly, we have added this information in lines 50-56 of the revised manuscript.
Additionally, as per your recommendation, we have cited less invasive alternatives for RHC treatment, including the reference 10.3390/jcm13164724 in lines 266-270.
We appreciate your suggestion to include the article https://doi.org/10.3390/nu15163573 in our study. However, upon careful consideration, we have determined that the content of this article is not directly relevant to the scope and focus of our research as it addresses urinary tract infections instead of radiation-induced cystitis. Therefore, we prefer not to cite it in our article. We believe that excluding this reference will help maintain the clarity and precision of our study.
We believe that these additions enhance the comprehensiveness of our study and effectively address your concerns.
Thank you once again for your insightful comments.
Best regards,